# Methoxy-Substituted Tyramine Derivatives Synthesis, Computational Studies and Tyrosinase Inhibitory Kinetics

**DOI:** 10.3390/molecules26092477

**Published:** 2021-04-23

**Authors:** Yasir Nazir, Hummera Rafique, Naghmana Kausar, Qamar Abbas, Zaman Ashraf, Pornchai Rachtanapun, Kittisak Jantanasakulwong, Warintorn Ruksiriwanich

**Affiliations:** 1Department of Pharmaceutical Sciences, Faculty of Pharmacy, Chiang Mai University, Chiang Mai 50200, Thailand; ynchem@yahoo.com; 2Department of Chemistry, Allama Iqbal Open University, Islamabad 44000, Pakistan; 3Department of Chemistry, University of Gujrat, Gujrat 50700, Pakistan; humera_rafique@uoe.edu.pk (H.R.); naghmana.kausar@uog.edu.pk (N.K.); 4Department of Biology, College of Science, University of Bahrain, Sakhir 32038, Bahrain; qamar.abbas.qau@gmail.com; 5Cluster of Agro Bio-Circular Green Industry (Agro BCG), Chiang Mai University, Chiang Mai 50100, Thailand; pornchai.r@cmu.ac.th (P.R.); Jantanasakulwong.k@gmail.com (K.J.); 6School of Agro-Industry, Faculty of Agro-Industry, Chiang Mai University, Chiang Mai 50100, Thailand; 7Cluster of Research and Development of Pharmaceutical and Natural Products Innovation for Human or Animal, Chiang Mai University, Chiang Mai 50200, Thailand

**Keywords:** tyramine derivatives, tyrosinase inhibitors, inhibitory activity, enzyme kinetics mechanism, computational studies

## Abstract

Targeting tyrosinase for melanogenesis disorders is an established strategy. Hydroxyl-substituted benzoic and cinnamic acid scaffolds were incorporated into new chemotypes that displayed in vitro inhibitory effects against mushroom and human tyrosinase for the purpose of identifying anti-melanogenic ingredients. The most active compound 2-((4-methoxyphenethyl)amino)-2-oxoethyl (*E*)-3-(2,4-dihydroxyphenyl) acrylate (Ph9), inhibited mushroom tyrosinase with an IC_50_ of 0.059 nM, while 2-((4-methoxyphenethyl)amino)-2-oxoethyl cinnamate (Ph6) had an IC_50_ of 2.1 nM compared to the positive control, kojic acid IC_50_ 16700 nM. Results of human tyrosinase inhibitory activity in A375 human melanoma cells showed that compound (Ph9) and Ph6 exhibited 94.6% and 92.2% inhibitory activity respectively while the positive control kojic acid showed 72.9% inhibition. Enzyme kinetics reflected a mixed type of inhibition for inhibitor Ph9 (*K_i_* 0.093 nM) and non-competitive inhibition for Ph6 (*K_i_* 2.3 nM) revealed from Lineweaver–Burk plots. In silico docking studies with mushroom tyrosinase (PDB ID:2Y9X) predicted possible binding modes in the catalytic site for these active compounds. Ph9 displayed no PAINS (pan-assay interference compounds) alerts. Our results showed that compound Ph9 is a potential candidate for further development of tyrosinase inhibitors.

## 1. Introduction

Tyrosinase is a copper containing metallo-enzymes and is widely distributed in nature, from bacteria to plants and humans. It facilitates the *o*-hydroxylation of monophenols to catechols following oxidation to quinones [1]. Its physiological function is the conversion l-tyrosine into dopaquinone, which constitutes melanin biosynthesis in melanosomes [2,3]. l-tyrosine or phosphorylated isomers of l-3,4-dihydroxyphenylalanine (l-DOPA) serve as key molecules in the regulation of the melanin biosynthesis [4,5,6]. The color of human skin, hair and eyes is determined by the distribution patterns of melanin in keratinocytes [7,8]. Several factors such as UV exposure, α-melanocyte-stimulating hormone and melanocortin 1 receptor have been reported to modulate melanogenesis [9,10]. The melanogenesis process is a potential cellular hazard and is limited to melanosomes in melanocytes, which synthesize pigments and transfer them to recipient cells [11]. The aberrant proliferation of melanocytes arises from a type of skin cancer called melanoma [12,13]. Malignant melanocytes tend to display up-regulated melanogenesis and malfunctioning melanosomes. Therefore, controlling a tyrosinase-dependent mechanism of melanogenesis might be the basis for a potential anti-melanoma therapy. Specific tyrosinase inhibitors like kojic acid, arbutin, azelaic acid, hydroquinone and electron-rich phenol and other chemical scaffolds like resorcinol [14,15,16], biaryl, β-phenyl-α,β-unsaturated carbonyl [17,18,19,20], and imidazolethione-containing inhibitors [21,22] have been tested in the search for their better tyrosinase ability to block the overproduction of melanin [23]. Thiosemicarbazone derivatives have been reported to possess mushroom tyrosinase inhibitory activity [24]. Inulavosin and its benzo-derivatives effectively inhibited the melanogenesis of the cells [25]. Arbutin and kojic acid were among the most popular specific tyrosinase inhibitors for topical application to the surface of the human skin because of beneficial effects on UV-irritated and dry skin by reducing the UV impact before serious side effects came to limit their human use [26]. Therefore, it is essential to find nontoxic skin-whitening agents for meeting the social needs in terms of safe cosmetic practices and medical hyperpigmentation treatments.

It should be highlighted that targeting tyrosinase to treat melanogenesis disorders is one of many possible approaches, due to the complex biochemical reaction involved in the melanin synthesis [27]. Tyrosinase polymerizes phenolic substrate through oxidative polymerization reactions by self-polymerization or cross-polymerization with other compounds [28]. Rafiq et al., 2019 and Nazir et al., 2020 reported that cinnamic acid and benzoic acid analogues possessing a substituted phenyl ring exhibited tyrosinase inhibitory activity [29,30]. It has already been well established that phenolic compounds such as caffeic acid, ferulic acid and *p*-coumaric acid play a vital role in tyrosinase inhibitory activity as the natural substrates tyrosine and l-DOPA also bear the phenolic hydroxyls [31]. Amides derived from coupling the coumaric acid and its derivatives with tyramine have been synthesized and have been studied in different therapeutic areas such as anticancer agents [32,33,34], antioxidants [35,36,37,38], and as selective *N*-methyl-d-aspartate (NMDA) receptor antagonists [39]. Roh et al., have reported melanogenesis-inhibitory effects of the naturally occurring *N*-feruloyl serotonin and *N*-(*p*-coumaroyl) serotonin, isolated from Safflower (*Carthamus tinctorius* L.) [40]. Hemp (*Cannabis sativa* L. var. *sativa*) contains a vast number of constituents and the terpenoids and flavonoids present in Hemp extract may be responsible for its tyrosinase inhibitory activity [41]. Tyrosinase inhibition activity of the Hemp leaf extract (IC_50_ 0.049 ± 0.02 mg/mL) is about 9.8 folds lower than standard kojic acid (IC_50_ 0.005 mg/mL) [42]. *Celastrus paniculatus* seed oil (CPSO) has been reported to exhibit more potent anti-tyrosinase activity (IC_50_ 0.04 mg/mL) than 2-hydroxypropylβ-cyclodextrin inclusion complex CPSO-HPβCD (IC_50_ 0.19 mg/mL) due to the presence of tocopherol [43]. The rice bran extract contains various saturated and unsaturated fatty acids such as palmitic, stearic, linolenic and oleic acids and their ester derivatives. The rice bran loaded in β-cyclodextrin at 1 mg/mL demonstrated the tyrosinase inhibitory activity of 1.25 fold of the control and 0.52 folds of the standard theophylline [44]. The key compounds, *p*-methylanisole, methyl salicylate, 2-phenyl ethyl alcohol, nerolidol, methyl cinnamate, 3-hydroxy-4-phenyl-2-butanoate, cinnamyl alcohol and 2-phenyl ethyl benzoate present in ethanol extract of the medicinal flower (*Mimusops elengi* L.) gave almost no tyrosinase inhibition activity (IC_50_ > 200 mg/mL) at all concentrations in comparison with kojic acid (IC_50_ 0.049 mg/mL) in the supercritical carbon dioxide fluid extraction (scCO_2_) [45,46]. Moreover, quercetagetin found in the ethyl acetate (EA) extract of flowers of African marigold (*Tagetes erecta* L.) exhibited tyrosinase inhibitory activity on l-tyrosine (IC_50_ 89.31 µg/mL), higher than α– and β–arbutins (IC_50_ 157.77 and 222.35 µg/mL) and slightly higher (IC_50_ 128.41 µg/mL) than ellagic acid (IC_50_ 151.1 µg/mL) when using l-DOPA as substrate [47]. In continuation of our research for finding potent tyrosinase inhibitors, we synthesized methoxy-substituted tyramine-based amides by coupling different hydroxyl-substituted benzoic and cinnamic acids to explore their anti-tyrosinase activity and to recognize how these synthesized compounds interact with the active site of the enzyme to search for more potent and novel compounds. These compounds were explored for their effect as inhibitors of human melanocyte-tyrosinase and mushroom tyrosinase and their ability as melanogenesis suppressers. The initial screening as an anti-melanogenic agent was carried out using mushroom and human (melanoma cells) tyrosinase. The in vitro tyrosinase inhibitory activity of the synthesized compounds (Ph1–Ph10) was evaluated both experimentally and in silico. Due to the absence of a high-resolution crystal structure of the human tyrosinase enzyme, mushroom tyrosinase has frequently served as a model to assess the inhibitory activity of compounds with the fact that both proteins share high sequence similarity in their active core region [48]. The in silico molecular docking of all the synthesized compounds (Ph1–Ph10) was performed against the mushroom tyrosinase crystal structure (PDB ID: 2Y9X) to determine the role of different functionalities in the formation of a ligand–protein complex. All these compounds (Ph1–Ph10) passed the PAINS (pan-assay interference compounds) filter, except for 2-((4-methoxyphenethyl)amino)-2-oxoethyl 3,4-dihydroxybenzoate (Ph4) [49]. The most potent derivatives, 2-((4-methoxyphenethyl)amino)-2-oxoethyl 3,5-dihydroxybenzoate (Ph5), 2-((4-methoxyphenethyl)amino)-2-oxoethyl cinnamate (Ph6) and 2-((4-methoxyphenethyl)amino)-2-oxoethyl (*E*)-3-(2,4-dihydroxyphenyl) acrylate (Ph9) were selected for further characterization of their biological effects in tyrosinase inhibitory kinetics.

## 2. Results and Discussion

### 2.1. Synthesis of 4-Methoxyphenyl Ethyl Chloroacetamide *(**2**)*

The intermediate 2-(4-methoxyphenyl)ethan-1-aminechloroacetate(2) was synthesized, purified by silica gel column and characterized by melting point and Fourier Transform Infrared (FTIR) spectroscopy. Melting point 99–101 °C; reaction time, 3 h; yield, 85%; Retardation factor (R_f_) 0.58 (*n*-hexane:ethyl acetate 3:1); FTIR, maximum wave number, the maximum velocity (*ν_max_*)cm^−1^:3345 (-NH), 2943 (sp^2^ C-H), 2834 (sp^3^ C-H), 1656 (C=O amide), 1589 (C=C aromatic), 1162 (C-O).

#### 2.1.1. In Vitro Tyrosinase Inhibition Assay

In vitro mushroom tyrosinase inhibitory assay of all the synthesized inhibitors (Ph1–Ph10) was performed and found that most of the title amide derivatives showed better tyrosinase inhibition than reference drug kojic acid. The hydroxylated amides with cinnamic acid moiety showed higher tyrosinase inhibition than those with benzoic acid analogues. The IC_50_ value of inhibitor Ph9 bearing 2,4-dihydroxyl-substituted cinnamic acid functionality was 0.000059 µM, whereas the inhibitor Ph5 bearing 3,5-dihydroxy-substituted benzoic acid moiety displayed an IC_50_ value of 0.231 µM which is excellent as compared with the standard drug kojic acid with IC_50_ 16.7 µM (Table 1). In vitro human tyrosinase inhibition assay showed that compound Ph9 exhibited 94.59% inhibition at 50 µg/mL (Table 1). This shows that the amide Ph9 has a greater potential to inhibit human tyrosinase compared to standard kojic acid (72.94% inhibitory capacity). Bioassay results revealed that the substitution of hydroxyl groups at cinnamic acid phenyl ring is the decisive factor for inhibitory activity. Hence, we propose that substitution of the hydroxyl group at *ortho, para* position on phenyl ring in Ph9 hampers the inhibitor to interact well within the enzymatic pocket of tyrosinase. The kinetics and docking studies facilitated us to explore the binding mode and amino acid residual interactions between the tyrosinase and synthesized inhibitors.

#### 2.1.2. Kinetics Mechanism

Among the synthesized Ph derivatives Ph5, Ph6 and Ph9 were selected to determine their inhibition type and inhibition constants on mushroom tyrosinase activity. The potential of these derivatives to inhibit the free enzyme and enzyme–substrate complex was determined in terms of enzyme inhibitor (EI) and enzyme–substrate-inhibitor (ESI) constants respectively. The kinetic studies of the enzyme by the Lineweaver–Burk plot of 1/*V* versus 1/[S] in the presence of different inhibitor concentrations gave a series of straight lines as shown in Figure 1A, Figure 2A and Figure 3A. The results of Figure 1A and Figure 3A showed that compounds Ph5 and Ph9 intersected within the second quadrant. The analysis showed that (*V_max_*) decreased with increasing Michaelis constant (*K_m_*) in the presence of increasing concentrations of compounds Ph5 and Ph9 respectively. This behavior of compounds Ph5 and Ph9 indicated that it inhibits tyrosinase by two different pathways; competitively forming enzyme inhibitor (EI) complex and interrupting enzyme–substrate-inhibitor (ESI) complex in a non-competitive manner. The secondary plots of slope versus concentration of compounds Ph5 and Ph9 showed EI dissociation constants (*K_i_*) Figure 1B and Figure 3B, while ESI dissociation constants (*K_i_*′) were shown by secondary plots of intercept versus concentration of compounds Ph5 and Ph9 Figure 1C and Figure 3C. A lower value of *K_i_* than *K_i_*′ pointed out stronger binding between enzyme and compounds Ph5 and Ph9 which suggested preferred competitive over non-competitive manners (Table 2). Meanwhile, in the case of compound Ph6, the Lineweaver–Burk plot gave a family of straight lines, all of which intersected at the same point on the x-axis Figure 2A. The analysis showed that 1/*V_max_* decreased to a new value while that of *K_m_* remains the same because of the increase in the concentrations of compound Ph6. This behavior indicated that compound Ph6 inhibits tyrosinase non-competitively to form an enzyme inhibitor (EI) complex. The secondary plot of slope against the concentration of Ph6 showed EI dissociation constant (*K_i_*) Figure 2B. It is worth mentioning here that the enzyme inhibitory kinetics were performed with a more efficient natural substrate l-DOPA (l-3,4-dihydroxyphenylalanine). Ph5 is a 3,5-dihydroxyl-substituted benzoic acid derivative and Ph9 is 2,4-dihydroxyl-substituted cinnamic acid derivative whereas, Ph6 is an unsubstituted cinnamic acid analogue. As the substrate l-DOPA bears 3,4-dihydroxy substitution, this could be the reason that Ph5 and Ph9 bearing hydroxyl substituent gave mixed-type inhibition, whereas Ph6 bearing no hydroxyl substituent was a non-competitive inhibitor.

#### 2.1.3. Molecular Docking Studies

Computational docking studies were undertaken with Maestro Schrodinger (11.2) to predict the most favorable binding mode of the compounds (Ph1–Ph10) inside the binding pocket of mushroom tyrosinase (PDB ID:2Y9X). The crystal structure of mushroom tyrosinase (PDB ID:2Y9X) containing a tropolone ligand in the enzymatic pocket, was retrieved from RCSB Protein Data Bank (RCSB PDB) (http://www.rcsb.org, accessed date 15 March 2021). To determine the amino acids of the protein involved in binding with the inhibitor, the selected poses of most active compounds Ph5, Ph6 and Ph9 were visualized.

Each inhibitor binds the protein in a different manner due to different substitution patterns. The reference inhibitor kojic acid interacts tyrosinase binuclear catalytic site with carbonyl group closer to cupper (400, 401) ions and side-chain hydroxyl is picking up a hydrogen bond (1.73 Å) with amino acid residue Met280. The best ligand binding poses of Ph5, Ph6 and Ph9 were selected based on binding energy (−5.339 Kcal/mol, −4.543 Kcal/mol and −7.93Kcal/mol) (Table 3), number and nature of amino acid units involved in the binding, type of interaction and inter-molecular distance between the ligand atoms and the amino acid residues. For the most active compound Ph9 (green in Figure 4d), both the *ortho* phenolic and *para* methoxy phenyl moieties are picking up H-bond interactions (2.93 Å, 3.04 Å) with side-chain residues Asn260 and Asn81 with bond lengths of 2.93 Å and 3.04 Å, respectively. The *para* phenolic group is interacting by Cu400 (3.68 Å) and Cu401 (3.10 Å) and this ring is further stabilized by *π*–*π* stacking with His85, His263 and His259. The amide hydrogen of Ph9 is predicted to form a hydrogen bond (3.36 Å) with the nitrogen of the neighboring His244. For the most active compound Ph5 from benzoic acid derivatives (Figure 4b), the *para* methoxy moiety is picking two strong H-bond interactions with side-chain residues Asn81 (2.07 Å) and His85 (2.31 Å) respectively and this ring is additionally stabilized by *π*–*π* stacking with His85. The amide hydrogen of the inhibitor Ph5 interacts through a hydrogen bond (2.40 Å) with the nitrogen of the neighboring His244. The large size and planner shape of Ph6 prevents it from entering the narrow binuclear copper-binding site and bounds to a shallow area at the surface of the enzyme pocket. However, a hydrogen bond (2.31 Å) is observed between the amide carbonyl and side-chain Asn81. Both *para* methoxy phenyl and cinnamic acid phenyl groups form *π*–*π* interaction with residues His85 and Phe192, respectively. The described binding mode would allow the substrate to enter the pocket, providing a useful non-competitive model. Our in vitro enzyme inhibitory assay showed a non-competitive inhibition for Ph6. The Ph6 is an unsubstituted cinnamic acid derivative and does not bear any hydroxyl substituent which is very important in ligand binding through hydrogen bonding, Coulomb and van der Waals interactions (Figure 4c). His259, His263, His296, His61, His85, His94 are conserved residues with Cu (400) and Cu (401) ions in the core region. The residues that played an important role in stabilizing the protein inhibitor complex are Asn260, Asn81, His85, His244 and Met280 and these findings are in accordance with the studies by Pintus et al. [50].

## 3. Materials and Methods

### 3.1. Chemistry

All chemicals used for the synthesis of compounds (Ph1–Ph10) were purchased from Sigma Aldrich (St. Louis, MO, USA). Melting points were determined using a Digimelt MPA 160, USA apparatus and are reported uncorrected. The FTIR spectra were recorded with Shimadzu FTIR–8400S spectrometer (Kyoto, Japan, υ, cm^−1^). Elemental Analysis (C, H) was carried out on a Flash 2000 series elemental analyzer with thermal conductivity detector (TCD) system (Thermo Flash 2000 elemental analyzer, Thermo Fisher Scientific Inc., Waltham, MA, USA) and results are with ±0.3%. The ^1^H NMR and ^13^C NMR spectra (DMSO-*d*_6_) were recorded using a Bruker 400 MHz spectrometer (Bruker, Freemont, CA, USA). Chemical shifts (*δ*) are reported in ppm downfield from the internal standard tetra methyl silane (TMS). The purity of the compounds was checked by thin layer chromatography (TLC) on silica gel plate using *n*-hexane and ethyl acetate as mobile phase. Mushroom tyrosinase, l-DOPA and 2-(4-methoxyphenyl)ethan-1-amine were purchased from Sigma Aldrich (St. Louis, MO, USA). Stock solutions of the reducing substrates were prepared in phosphate buffer (20 mM, pH 6.8).

### 3.2. General Procedure for the Synthesis of Title Compounds (Ph1–Ph5) and (Ph6–Ph10)

The intermediate 2-(4-methoxyphenyl)ethan-1-aminechloroacetate (**2**) was synthesized by following the Sidhu et al., method with some modifications [51]. The 2-(4-methoxyphenyl) ethan-1-amine (**1**) (0.01 mol) was dissolved in anhydrous dichloromethane (DCM) (25 mL) and triethylamine (C_2_H_5_)_3_N (0.01 mol) was added slowly. The resulting solution was then cooled in an ice salt mixture to 0 to −5 °C. The chloroacetyl chloride (0.01 mol) in dry DCM was added drop-wise to the reaction mixture with constant stirring over a period of 1h maintaining the temperature constant. The reaction mixture was then stirred at room temperature for a further 5 h, washed with 5% HCl, and 5% sodium hydroxide solution. The organic layer was washed with saturated aqueous NaCl, dried over anhydrous magnesium sulfate, filtered and the solvent was removed under reduced pressure. The crude product was purified by silica gel column to afford the corresponding 2-(4-methoxyphenyl)ethan-1-aminechloroacetate (**2**).

The substituted benzoic acids (3a–e) (0.01 mol), triethylamine (0.01 mol) and potassium iodide (KI) (0.01 mol) were mixed in dimethyl formamide (DMF) (25 mL) and stirred at room temperature. The 2-(4-methoxyphenyl)ethan-1-amine chloroacetate intermediate (2) was then added to the reaction mixture slowly and stirred the reaction mixture at the same temperature overnight (Scheme 1). The final products (Ph1–Ph5) were then extracted with ethyl acetate (3 × 25 mL). The combined ethyl acetate layer was then washed with 5% HCl, 5% sodium carbonate and finally with brine solution. The organic layer was dried over anhydrous magnesium sulfate, filtered and the solvent was removed under reduced pressure to afford the crude products (Ph1–Ph5). The title compounds (Ph1–Ph5) were purified by silica gel column chromatography (*n*-hexane: ethyl acetate 3:1). The same procedure was used for the preparation of compounds (Ph6–Ph10) Scheme 2.

### 3.3. Spectral Characterization of Synthesized Compounds (Ph1–Ph10)

***2-((4-methoxyphenethyl)amino)-2-oxoethyl 3-hydroxybenzoate* (Ph1)** solid; melting point, 86–88 °C; R_f_ 0.54 (*n*-hexane:ethyl acetate 3:1); FTIR ν_max_ cm^−1^: 3378 (O-H), 3245 (-NH), 2967 (sp^2^ C-H), 2856 (sp^3^ C-H), 1729 (C=O ester), 1645 (C=O amide), 1592 (C=C aromatic), 1187 (C-O, ester); ^1^H NMR (400 MHz, DMSO, ppm): *δ*9.86 (s, 1H,-OH), 8.27 (t, *J* = 4.4 Hz, 1H,-NH), 7.46 (dd *J* = 6.4, 8 Hz, 1H), 7.42 (t, *J* = 4 Hz, 1H), 7.13 (d, *J* = 8.8 Hz, 2H), 7.07 (dd, *J* = 6.4, 8 Hz, 1H), 6.86 (d, *J* = 8.8 Hz, 2H), 6.83 (d, *J* = 4 Hz, 1H), 4.04 (s, 2H,-CH_2_), 3.72 (s, 3H,-OCH_3_), 3.28 (d,d, *J* = 6, 7.6 Hz, 2H,-CH_2_), 2.67 (t, *J* = 7.6 Hz, 2H,-CH_2_), ^13^C NMR (100 MHz, DMSO, ppm); *δ*167.95 (C=O, amide), 166.92 (C=O, ester), 158.19, 157.95, 131.63, 131.54, 131.49, 130.96, 120.98, 120.65, 116.43, 114.24, 63.39 (-CH_2_), 55.45 (-OCH_3_), 41.39 (-CH_2_), 34.44 (-CH_2_) (Appendix A). Elem. Anal. Calc. for C18H19NO5; C, 65.64; H, 5.82; Found; C, 65.66; H, 5.84.

***2-((4-methoxyphenethyl)amino)-2-oxoethyl 4-hydroxybenzoate*****(Ph2)** solid; melting point, 93–95 °C; R_f_ 0.53 (*n*-hexane:ethyl acetate 3:1); FTIR ν_max_ cm^−^^1^: 3364 (O-H), 3278 (-NH), 2956 (sp^2^ C-H), 2867 (sp^3^ C-H), 1723 (C=O ester), 1652 (C=O amide), 1597 (C=C aromatic), 1178 (C-O, ester); ^1^H NMR (400 MHz, DMSO, ppm): *δ*10.39 (s, 1H,-OH), 8.26 (t, *J* = 4.4 Hz, 1H,-NH), 7.88 (d, *J* = 4.8 Hz, 2H), 7.13 (d, *J* = 8.4 Hz, 2H6.88 (d, *J* = 8.4 Hz, 2H), 6.84 (d, *J* = 4.8 Hz, 2H), 4.03 (s, 2H,-CH_2_), 3.72 (s, 3H,-OCH_3_), 3.28 (d,d, *J* = 6, 7.6 Hz, 2H,-CH_2_), 2.67 (t, *J* = 7.6 Hz, 2H,-CH_2_), ^13^C NMR (100MHz, DMSO, ppm); *δ*166.23 (C=O, amide), 165.48, 162.67 (C=O, ester), 158.16, 132.32, 131.64, 130.06, 120.31, 115.75, 114.24, 63.05 (-CH_2_), 55.45 (-OCH_3_), 43.1 (-CH_2_), 34.43 (-CH_2_) (Appendix A). Elem. Anal. Calc. for C18H19NO5; C, 65.64; H, 5.82; Found; C, 65.62; H, 5.80.

***2-((4-methoxyphenethyl)amino)-2-oxoethyl 2,4-dihydroxybenzoate*****(Ph3)** solid; melting point, 122–124 °C; R_f_ 0.51 (*n*-hexane:ethyl acetate 3:1); FTIR ν_max_ cm^−^^1^: 3343 (O-H), 3276 (-NH), 2944 (sp^2^ C-H), 2865 (sp^3^ C-H), 1725 (C=O ester), 1639 (C=O amide), 1601 (C=C aromatic), 1184 (C-O, ester); ^1^H NMR (400 MHz, DMSO, ppm): *δ*10.53 (s, 1H,-OH), 8.27 (t, *J* = 4.4 Hz, 1H,-NH), 7.74 (d, *J* = 8.8 Hz, 1H), 7.12 (d, *J* = 5.2 Hz, 2H), 6.84 (d, *J* = 5.2 Hz, 2H), 6.83 (dd, *J* = 5.2, 2.0 Hz, 1H), 6.32 (d, *J* = 2.0 Hz, 1H), 4.03 (s, 2H,-CH_2_), 3.72 (s, 3H,-OCH_3_), 3.28 (d,d, *J* = 6, 7.6 Hz, 2H,-CH_2_), 2.67 (t, *J* = 7.6 Hz, 2H,-CH_2_), ^13^C NMR (100MHz, DMSO, ppm); *δ*168.68 (C=O, amide), 166.79 (C=O, ester), 166.23, 164.86, 158.19, 131.54, 131.49, 131.24, 130.09, 130.06, 108.79, 104.42, 63.37 (-CH_2_), 55.45 (-OCH_3_), 40.62 (-CH_2_), 34.43(-CH_2_) (Appendix A). Elem. Anal. Calc. for C18H19NO6; C, 62.60; H, 5.55; Found; C, 62.62; H, 5.57.

***2-((4-methoxyphenethyl)amino)-2-oxoethyl 3,4-dihydroxybenzoate*****(Ph4)** solid; melting point, 112–114 °C; R_f_ 0.55 (*n*-hexane:ethyl acetate 3:1); FTIR ν_max_ cm^−^^1^: 3367 (O-H), 3249 (-NH), 2923 (sp^2^ C-H), 2865 (sp^3^ C-H), 1728 (C=O ester), 1642 (C=O amide), 1589 (C=C aromatic), 1159 (C-O, ester); ^1^H NMR (400 MHz, DMSO, ppm): *δ*9.88 (s, 1H, H-7, -OH), 7.98 (d, *J* = 5.2 Hz, 2H), 7.75 (d, *J* = 5.2 Hz, 2H), 7.43 (d, *J* = 2.0 Hz, 1H), 7.40 (dd, *J* = 8.4, 2.0 Hz, 1H), 6.86 (d, *J* = 8.0 Hz, 1H), 4.04 (s, 2H,-CH_2_), 3.72 (s, 3H,-OCH_3_), 3.28 (d,d, *J* = 6, 7.6 Hz, 2H,-CH_2_), 2.67 (t, *J* = 7.6 Hz, 2H,-CH_2_), ^13^C NMR (100 MHz, DMSO, ppm); *δ*166.78 (C=O, amide), 165.8 (C=O, ester), 151.26, 145.6, 143.32132.42, 130.01, 122.69, 120.39, 118.96, 117.01, 115.85, 63.26 (-CH_2_), 55.45 (-OCH_3_), 40.62 (-CH_2_), 34.43 (-CH_2_) (Appendix A). Elem. Anal. Calc. for C18H19NO6; C, 62.60; H, 5.55; Found; C, 62.59; H, 5.54.

***2-((4-methoxyphenethyl)amino)-2-oxoethyl 3,5-dihydroxybenzoate*****(Ph5)** solid; melting point, 132–134 °C; R_f_ 0.53 (*n*-hexane:ethyl acetate 3:1); FTIR ν_max_ cm^−^^1^: 3354 (O-H), 3246 (-NH), 2934 (sp^2^ C-H), 2843 (sp^3^ C-H), 1732 (C=O ester), 1640 (C=O amide), 1592 (C=C aromatic), 1160 (C-O, ester); ^1^H NMR (400 MHz, DMSO, ppm): *δ*9.67 (s, 2H, -OH), 8.25 (t, *J* = 4.4 Hz, 1H,-NH), 7.14 (d, *J* = 2.8 Hz, 2H), 6.88 (s, 2H), 6.84 (d, *J* = 2.8 Hz, 2H), 6.48 (t, *J* = 2 Hz, 1H), 4.06 (s, 2H, -CH_2_), 3.72 (s, 3H,-OCH_3_), 3.28 (d,d, *J* = 6, 7.6 Hz, 2H,-CH_2_), 2.67 (t, *J* = 7.6 Hz, 2H,-CH_2_), ^13^C NMR (100 MHz, DMSO, ppm); *δ*167.96 (C=O, amide), 165.81 (C=O, ester), 158.19, 158.17, 131.49, 130.06, 129.75, 114.24, 107.94, 107.85, 63.32 (-CH_2_), 55.44 (-OCH_3_), 40.84 (-CH_2_), 34.43 (-CH_2_) (Appendix A). Elem. Anal. Calc. for C18H19NO6; C, 62.60; H, 5.55; Found; C, 62.61; H, 5.56.

***2-((4-methoxyphenethyl)amino)-2-oxoethyl cinnamate*****(Ph6)** solid; melting point, 103–105 °C; R_f_ 0.61 (*n*-hexane:ethyl acetate 3:1);FTIR ν_max_ cm^−^^1^: 3297 (-NH), 2932 (sp^2^ C-H), 2843 (sp^3^ C-H), 1726 (C=O ester), 1651 (C=O amide), 1597 (C=C aromatic), 1186 (C-O, ester);^1^H NMR (400 MHz, DMSO, ppm): *δ*8.27 (t, *J* = 4.4 Hz, 1H,-NH), 7.76 (d, *J* = 2.8 Hz, 2H), 7.73 (d, *J* = 16 Hz, 1H), 7.45 (d, *J* = 4.4 Hz, 2H), 7.12 (d, *J* = 8.4 Hz, 2H), 6.86 (d, *J* = 4.4 Hz, 2H), 6.84 (d, *J* = 8.4 Hz, 2H), 6.72 (d, *J* = 16 Hz, 1H), 4.03 (s, 2H,-CH_2_), 3.72 (s, 3H,-OCH_3_), 3.28 (d,d, *J* = 6, 7.6 Hz, 2H,-CH_2_), 2.67 (t, *J* = 7.6 Hz, 2H,-CH_2_), ^13^C NMR (100 MHz, DMSO, ppm); *δ*167.03 (C=O, amide), 166.23 (C=O, ester), 158.19, 145.57, 134.45, 131.62, 131.49, 131.10, 130.06, 128.88, 118.09, 114.24, 62.85 (-CH_2_), 55.45 (-OCH_3_), 40.79 (-CH_2_), 34.43 (-CH_2_) (Appendix A). Elem. Anal. Calc. for C20H21NO4; C, 70.78; H, 6.24; Found; C, 70.80; H, 6.26.

***2-((4-methoxyphenethyl)amino)-2-oxoethyl (E)-3-(2-hydroxyphenyl)*****acrylate (Ph7)** solid; melting point, 111–113 °C; R_f_ 0.55 (*n*-hexane:ethyl acetate 3:1);FTIR ν_max_ cm^−^^1^: 3343 (O-H), 3267 (-NH), 2949 (sp^2^ C-H), 2845 (sp^3^ C-H), 1731 (C=O ester), 1654 (C=O amide), 1593 (C=C aromatic), 1186 (C-O, ester); ^1^H NMR (400 MHz, DMSO, ppm): *δ*10. 38 (s,1H, -OH), 8.12 (t, *J* = 5.6 Hz, 1H, -NH), 7.93 (d, *J* = 16 Hz, 1H), 7.63 (d, *J* = 7.6 Hz, 1H), 7.26 (t, *J* = 7.2 Hz, 1H), 7.13 (d, *J* = 8.4 Hz, 2H), 6.94 (d, *J*= 7.6 Hz, 1H), 6.87 (t, *J* = 7.6 Hz, 1H), 6.84 (d, *J* = 8.4 Hz, 2H), 6.7 (d, *J* = 16 Hz, 1H), 4.04 (s, 2H,-CH_2_), 3.72 (s, 3H,-OCH_3_), 3.28 (d,d, *J* = 6, 7.6 Hz, 2H, -CH_2_), 2.67 (t, *J* = 7.6 Hz, 2H -CH_2_), ^13^C NMR (100 MHz, DMSO, ppm); *δ*167.17 (C=O, amide), 166.64 (C=O, ester), 158.19, 157.40, 141.28, 132.34, 130.06, 129.75, 129.52, 121.12, 119.91, 117.18, 116.69, 114.24, 62.74 (-CH_2_), 55.4 (-OCH_3_), 40.8 (-CH_2_), 34.44 (-CH_2_) (Appendix A). Elem. Anal. Calc. for C20H21NO5; C, 67.59; H, 5.96; Found; C, 67.61; H, 5.98.

***2-((4-methoxyphenethyl)amino)-2-oxoethyl(E)-3-(4-hydroxyphenyl)*****acrylate (Ph8)** solid; melting point, 119–121 °C; R_f_ 0.53 (*n*-hexane:ethyl acetate 3:1); FTIR ν_max_ cm^−^^1^: 3371 (O-H), 3267 (-NH), 2976 (sp^2^ C-H), 2859 (sp^3^ C-H), 1733 (C=O ester), 1655 (C=O amide), 1598 (C=C aromatic), 1177 (C-O, ester); ^1^H NMR (400 MHz, DMSO, ppm): δ10.06 (s, 1H,-OH), 8.26 (t, *J* = 5.2 Hz, 1H,-NH), 7.63 (d, *J* = 16 Hz, 1H), 7.59 (d, *J* = 2 Hz, 2H), 7.14 (d, *J* = 3.2 Hz, 2H), 6.86 (d, *J* = 3.2 Hz, 2H), 6.81 (d, *J* = 2 Hz, 2H), 6.46 (d, *J* = 16 Hz, 1H), 4.03 (s, 2H,-CH_2_), 3.72 (s, 3H,-OCH_3_), 3.27 (d,d *J* = 6,7.6 Hz, 2H, -CH_2_), 2.66 (t, *J* = 7.6 Hz, 2H, -CH_2_), ^13^C NMR (100MHz, DMSO, ppm); δ167.18 (C=O, amide), 166.23 (C=O, ester), 160.45, 158.19, 145.81, 131.64, 130.87, 130.06, 125.52, 116.29, 114.24, 114.14, 62.65 (-CH_2_), 55.42 (-OCH_3_), 41.27 (-CH_2_), 34.67 (-CH_2_) (Appendix A). Elem. Anal. Calc. for C20H21NO5; C, 67.59; H, 5.96; Found; C, 67.58; H, 5.95.

***2-((4-methoxyphenethyl)amino)-2-oxoethyl(E)-3-(2,4-dihydroxyphenyl)acrylate*****(Ph9**) solid; melting point, 175–177 °C; R_f_ 0.51 (*n*-hexane:ethyl acetate 3:1); FTIR ν_max_ cm^−^^1^: 3365 (O-H), 3276 (-NH), 2959 (sp^2^ C-H), 2869 (sp^3^ C-H), 1730 (C=O ester), 1647 (C=O amide), 1603 (C=C aromatic), 1178 (C-O, ester); ^1^H NMR (400 MHz, DMSO, ppm): *δ*10.22 (s, 1H, -OH), 9.93 (s, 1H,-OH), 8.27 (t, *J* = 5.2 Hz, 1H,-NH), 7.82 (d, *J* = 16 Hz, 1H), 7.44 (d, *J* = 8.8 Hz, 1H), 7.14 (d, *J* = 2.4, 2H), 6.86 (d, *J* = 2.4 Hz, 2H), 6.47 (d, *J* = 16 Hz, 1H), 6.39 (d, *J* = 2 Hz, 1H), 6.31 (d, *J* = 2 Hz, 1H), 4.04 (s, 2H,-CH2), 3.72 (s,3H,-OCH_3_), 3.27 (d,d *J* = 6,7.6 Hz, 2H,-CH_2_), 2.67 (t, *J* = 7.6 Hz, 2H,-CH_2_), ^13^C NMR (100MHz, DMSO, ppm); δ167.36 (C=O, amide), 166.24 (C=O, ester), 161.60, 159.18, 158.19, 158.16, 141.72, 131.66, 131.07, 129.98, 129.75, 114.24, 112.91, 108.34, 62.52 (-CH_2_), 55.44 (-OCH_3_), 43.10 (-CH_2_), 34.68 (-CH_2_) (Appendix A). Elem. Anal. Calc. for C20H21NO6; C, 64.68; H, 5.70; Found; C, 64.70; H, 5.72.

***2-((4-methoxyphenethyl)amino)-2-oxoethyl (E)-3-(4-chlorophenyl)acrylate*****(Ph10)** solid; melting point, 107–109 °C; R_f_ 0.58 (*n*-hexane:ethyl acetate 3:1); FTIR ν_max_ cm^−^^1^: 3278 (-NH), 2956 (sp^2^ C-H), 2887(sp^3^ C-H), 1727 (C=O ester), 1657 (C=O amide), 1598 (C=C aromatic), 1180 (C-O, ester);^1^H NMR (400 MHz, DMSO, ppm): δ8.27 (t, *J* = 5.2 Hz, 1H -NH), 7.8 (d, *J* = 8.4 Hz, 2H), 7.72 (d, *J* = 16 Hz, 1H), 7.52 (d, *J* = 8.4 Hz, 2H), 7.12 (d, *J* = 2 Hz, 2H), 6.87 (d, *J*= 2 Hz, 2H), 6.74 (d, *J* = 16 Hz, 1H), 4.03 (s, 2H, -CH_2_), 3.72(s, 3H, -OCH_3_), 3.28 (d,d *J* = 7.6,6.0 Hz, 2H,-CH_2_), 2.67 (t, *J* = 7.6 Hz, 2H, -CH_2_), ^13^C NMR (100 MHz, DMSO, ppm); δ166.98 (C=O, amide), 166.23 (C=O, ester), 158.17, 144.14, 135.60, 133.43, 131.49, 130.61, 130.06, 129.49, 118.94, 114.24, 62.89 (-CH_2_), 55.42 (-OCH_3_), 41.27 (-CH_2_), 34.43 (-CH_2_) (Appendix A). Elem. Anal. Calc. for C20H20ClNO4; C, 64.26; H, 5.39; Found; C, 64.24; H, 5.37.

### 3.4. Mushroom Tyrosinase Inhibition Assay

In vitro mushroom tyrosinase (Sigma Aldrich, St. Louis, MO, USA) inhibition assay was performed, following our already reported method [52]. In detail, 140 µL of phosphate buffer (20 mM, pH 6.8), 20 µL of mushroom tyrosinase (30 U/mL) and 20 µL of the inhibitor solution were placed in the wells of a 96-well microplate. After pre-incubation for 10 min at room temperature, 20 µL of l-DOPA (3,4-dihydroxyphenylalanine, Sigma Aldrich, St. Louis, MO, USA) (0.85 mM) was added and the assay plate was further incubated at 25 °C for 20 min. Afterward the absorbance of dopachrome was measured at 475 nm using a microplate reader (OPTIMax, Tunable, Sunnyvale, CA, USA). All of the compounds were dissolved in DMSO and blank tests were performed with solvent DMSO. The IC_50_ values were reported after subtracting the value of DMSO. Kojic acid was used as a reference inhibitor (positive control). The amount of inhibition by the test compounds was expressed as the percentage of concentration necessary to achieve 50% inhibition (IC_50_). Each concentration was analyzed in three independent experiments. The IC_50_ values were determined by the data analysis and graphing software Origin 8.6, 64-bit.

The tyrosinase %age inhibition was determined by using the following formula:Inhibition (%) = [(B − S)/B] × 100(1)

Here, the absorbance of the blank and inhibitor are represented by B and S respectively.

### 3.5. Human Tyrosinase Inhibition Assay

#### 3.5.1. Cell Culture and Preparation of Tyrosinase

A375 human melanoma cells were obtained from American Type Culture Collection (ATCC; Rockville, MD, USA). A375 cells were grown in Dulbecco’s modified Eagle’s medium (DMEM; Invitrogen, Burlington, ON, Canada) supplemented with L-glutamine, 10% (*v*/*v*) fetal bovine serum (FBS; Invitrogen), 50 µg/mL streptomycin (Sigma Aldrich, St. Louis, MO, USA), 50 units/mL penicillin (Sigma Aldrich) and supplemented with 200 µM of l-tyrosine for tyrosinase induction. Cell cultures were incubated at 37 °C, in a humidified atmosphere of 5% CO_2_. Cells were scraped out from the tissue culture plate with phosphate-buffer saline (PBS) and were homogenized at 4 °C in PBS. The homogenate was centrifuged at 1000× *g* for 10 min. The precipitate was sonicated in PBS on ice and the mixture was centrifuged at 10,000× *g* for 30 min. The supernatant containing tyrosinase was used for the measurement of the inhibitory effects.

#### 3.5.2. Human Tyrosinase Inhibition Assay

The tyrosinase inhibitory activity of the synthesized amides (Ph1–Ph10) was determined by following the already reported method with few modifications [53,54]. The assay reaction mixture (200 µL) contained 3.3 mM l-DOPA in 0.33 M phosphate buffer (pH 7.0) and the enzyme in the presence and absence of inhibitors. Fifteen or twenty units of tyrosinase were used to determine the % inhibition. The reaction mixture was incubated at 37 °C for 10 min and the absorbance was recorded at 475 nm using a microplate reader (OPTI Max, Tunable). One unit of the enzyme was defined as the amount of enzyme which increases the absorbance value by 0.001 at 475 per minute under the same conditions as described above.

### 3.6. Kinetic Analysis of the Inhibition of Tyrosinase

A series of experiments were performed to calculate the inhibitory kinetics of compounds Ph5, Ph6 and Ph9 by following the already reported method [55]. The compounds concentrations are: 0, 0.2 and 0.4 µM for Ph5; 0, 0.001 and 0.002 µM for Ph6; 0, 0.00003, 0.00006, and 0.00012 µM for Ph9. The concentration of substrate l-DOPA was between 0.125 to 2 mM in all kinetic studies. Pre-incubation time and measurement time were the same as discussed in mushroom tyrosinase inhibition assay protocol. Maximal initial velocity was determined from initial linear portion of absorbance up to five minutes after addition of enzyme at a 30 s interval. The inhibition type on the enzyme was determined by Lineweaver–Burk plots of inverse velocities (1/*V*) versus the inverse of substrate concentration 1/[S] mM^−1^. The EI-dissociation constant *K_i_* was determined by the secondary plot of 1/*V* versus inhibitor concentrations while ESI-dissociation constant *K_i_*′ was determined by intercept versus inhibitor concentrations.

### 3.7. Molecular Docking Studies

We studied all the synthesized compounds (Ph1–Ph10) using molecular docking simulations to determine possible binding orientations of the compounds in the crystal structure of the mushroom tyrosinase. For this purpose, we used the crystal structure of the *Agaricus bisporus* tyrosinase (PDB:2Y9X) from the RCSB Protein Data Bank (RCSB PDB). 2Y9X is a crystal structure of mushroom tyrosinase complex with tropolone. Tropolone was removed from the active site and all compounds (Ph1–Ph10) were docked in the active site. All ligands were prepared by LigPrep (Schrödinger) in their neutral form and their conformation optimized in the OPLS-3 force field. The protein structure was prepared by using one of the four monomers from the PDB entry and using the Protein Preparation (Schrödinger) for adding hydrogens and by removing all water molecules beyond 5.0 Å and setting protonation states appropriate for pH 7. The receptor grid box was defined as 20 Å. Docking was performed with Glide (Schrödinger) using extra precision (XP) with default settings and glide scoring function, reporting the 15 top ranked poses for each ligand. Visual inspection of the binding pose and generation of figure was done with Maestro (Schrödinger) [50].

### 3.8. Structure–Activity Relationship (SAR)

In the present study novel tyrosinase inhibitors (Ph1–Ph10) containing amide and ester functionalities were synthesized and evaluated for their tyrosinase inhibitory potential against mushroom and melanoma cells human tyrosinase. Mushroom tyrosinase being commercially available has served almost in all studies conducted on tyrosinase inhibition so far. Phenolics are the largest groups in tyrosinase inhibitors and several polyphenols are accepted as substrates by tyrosinase, depending upon the presence and position of substituent [56]. Cinnamic acid has been reported to possess broad physiological actions involving tyrosinase inhibitory activity [57]. The synthesized compounds (Ph6–Ph10) are all cinnamate esters/amides but differ in the substitution pattern on the cinnamate phenyl ring. Compound (Ph6) bears an unsubstituted cinnamic acid moiety while compound (Ph7), (Ph8) and (Ph9) possess 2-hydroxy, 4-hydroxy and 2,4-dihydroxy-substituted cinnamic acid component, respectively. The tyrosinase inhibitory activity of 2,4-dihydroxy cinnamic acid is already reported [58] and its incorporation as an ester/amide in compound Ph9 resulted in an increased inhibitory potential. The replacement of hydrogen from the para position of the cinnamic acid phenyl ring in Ph6 with a 2,4-dihydroxyls in Ph9 led to a significant increase in tyrosinase inhibitory activity (Table 1).

It was found that inhibitor Ph9 bearing dihydroxy-substituted cinnamic acid moiety is the most promising compound with an inhibitory potential (IC_50_ 0.00006 µM) much better than the standard kojic acid (IC_50_ 16.7 µM). The enhanced inhibitory effect of Ph9 is potentially due to the presence of two hydroxyl groups at *ortho* and *para* positions of phenyl ring of cinnamic acid moiety. Whereas the Ph6 bearing an unsubstituted cinnamic acid moiety also displayed an excellent enzyme inhibitory activity (IC_50_ 0.00204 µM). The inhibitor Ph5 with two hydroxyls at 3,5 positions of benzoic acid moiety also exhibited excellent tyrosinase inhibition (IC_50_ 0.231 µM) (Figure 5). The bioassay results proved that derivatives with hydroxy-substituted cinnamic acid moiety (Ph7-Ph9) displayed greater tyrosinase inhibitory activity compared to hydroxy-substituted benzoic acid derivatives (Ph1-Ph5). It has already been demonstrated that phenolic hydroxyls play a key role in tyrosinase inhibition activity as the natural substrates l-tyrosine and l-DOPA also bear the phenolic hydroxyls [59]. Here, we report that the *ortho* and *para*-hydroxyls of Ph9 seem to perform a similar role in tyrosinase inhibitory activity.

## 4. Conclusions

The mushroom and human tyrosinase inhibitors Ph1–Ph10 with excellent tyrosinase inhibitory potential compared to the reference kojic acid are presented in the current work. A series of compounds Ph1–Ph10 containing ester/amide functionalities were synthesized and evaluated to check their inhibitory potential against tyrosinase enzyme. Simple nucleophilic substitution synthetic routes are adopted to synthesize the desired compounds in good and excellent yield (71–84%). The mushroom (IC_50_) and human tyrosinase (% age) enzyme inhibitory activities showed that the compound Ph9 (IC_50_ 0.00005950 µM, 94.59%) with 2,4-dihydroxycinnamic acid moiety, Ph6 (IC_50_ 0.00204 µM, 92.24%) bearing unsubstituted cinnamic acid and inhibitor Ph5 (IC_50_ 0.231 µM, 64.8%) containing 3,5-dihydroxybenzoic acid unit showed excellent enzyme inhibitory potential, much better than the standard kojic acid (IC_50_ 16.7 µM, 72.94%). The presence of *ortho* and *para* hydroxyl substituents on phenyl ring and *p*-methoxy-substituted benzene ring on the other side of Ph9 played a vital role in enhanced tyrosinase inhibitory activity. Kinetic mechanism reflected the mixed type of inhibition for inhibitors Ph9 (*K_i_* 0.000093 µM), Ph5 (*K_i_* 0.219 µM) and non-competitive inhibition for Ph6 (*K_i_* 0.0023 µM) revealed from Lineweaver–Burk plots. In silico molecular docking studies showed that the predicted binding affinity of the synthesized compound Ph9 is excellent with a binding energy −7.93 Kcal/mol. It can be concluded from our results that the inhibitor Ph9 is highly potent and may be a model structure for the development of novel and potent melanogenic inhibitors.

## Data Availability

The data presented in this study are available in insert article and Appendix A here.

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
