# Peer review of "Methoxy-Substituted Tyramine Derivatives Synthesis, Computational Studies and Tyrosinase Inhibitory Kinetics"

_molecules, 2021, doi:10.3390/molecules26092477_

Round 1

Reviewer 1 Report

The studies of the inhibitors of tyrozinase presented in a paper are very interesting and have potential - especially in the case of the best inhibitors.

Maybe some results of cytotoxity of the investigated compounds would be also interesting.

The autors should also make alittle deeper litearture study - for example - look on papers of Latajka group.

I recommand this paper for publication in Molecules.

Author Response

Response to reviewer’s comments:

Manuscript ID: Molecules-1171850

Type of Manuscript: Article

Title: Methoxy substituted tyramine derivatives synthesis, computational studies and tyrosinase inhibitory kinetics

Authors: Y. Nazir et al.

REVIEWER 1

  1. The studies of the inhibitors of tyrosinase presented in a paper are very interesting and have potential - especially in the case of the best inhibitors. Maybe some results of cytotoxicity of the investigated compounds would be also interesting. The authors should also make a little deeper literature study - for example - look on papers of Latajka group.

Response: In fact, this is a very nice suggestion to check the cytotoxicity of the synthesized compounds (Ph1-Ph10) but, unfortunately the cytotoxicity test in cell lines is not available in our lab. The cytotoxicity assay will be performed in future studies. Latajka group has published many interesting articles on tyrosinase inhibitors mostly working on thiosemicarbazones and triazoles and was helpful in writing this manuscript. This sentence was added “Thiosemicarbazone derivatives have been reported to possess mushroom tyrosinase inhibitory activity [24]. Please see on page 2, lines 54-55.

Reference:

  1. Hałdys, K.; Goldeman, W.; Anger-Góra, N.; Rossowska, J.; Latajka, R. Monosubstituted acetophenone thiosemicarbazones as potent inhibitors of tyrosinase: synthesis, inhibitory studies, and molecular docking. Pharmaceuticals 2021, 14, 74. doi.org/10.3390/ph14010074.

  1. English language and style are fine/minor spell check required.

Response: Spellings are thoroughly checked and revised as per reviewer recommendation.

Reviewer 2 Report

To contribute to the development of potent tyrosinase inhibitors (as novel anti-melanogenic agents), the authors synthesized methoxy substituted tyramine-based amides through coupling of hydroxyl substituted benzoate and cinnamate and further performed in vitro and preliminary in silico evaluations to understand the nature of interaction(s) of the synthesized compounds with active site residues of the study enzyme. Judging by the results of the molecular docking and the initial screening of the compounds in A375 human melanoma cells and mushroom tyrosinase inhibition assay, Ph5, Ph6 and Ph9 were selected as the most potent of the resulting 10 derivatives (Ph1-Ph10), and further characterized for their inhibition kinetics. Of these, Ph9 was concluded as a potential candidate for further development as novel tyrosinase inhibitors.

The following concerns must be addressed:

  1. It was stated that tyrosinase was non-competitively inhibited by Ph6 as revealed by Lineweaver-Burk plot. However, for this to be so, Ph6 should interact with the non-competitive/allosteric domain of the enzyme rather than the active site where docking was reported to have been performed. Hence, how do the authors achieved the docking of Ph6 at the active site contrary to the enzyme inhibition kinetics that was predicted to be non-competitive? This is a key question/concern requiring detailed explanation to support that that the in silico evaluation for Ph6 really makes scientific sense and of significance to the study.
  2. Include the interaction plots for Ph6 and Kojic acid as done for Ph5 and Ph9 (Figure 5) and compare the plots for Ph5, Ph6 and Ph9 with that of Kojic acid, being the reference standard used and clearly identify the conserved catalytic residues crucial for tyrosinase activity in consultation with other published works. A mere descriptive manner of presentation as currently done in this study is unacceptable.
  3. PDB: 2Y9X is a crystal structure of tyrosinase in complex with an inhibitor. Define the amino sequence of this structure and include information on how the structure was treated regarding the presence of the inhibitor.
  4. Provide information on the source and purity of the parent/starting compounds for the synthetic routes explored in the study.

Author Response

Response to reviewer’s comments:

Manuscript ID: Molecules-1171850

Type of Manuscript: Article

Title: Methoxy substituted tyramine derivatives synthesis, computational studies and tyrosinase inhibitory kinetics

Authors: Y. Nazir et al.

REVIEWER 2

  1. It was stated that tyrosinase was non-competitively inhibited by Ph6 as revealed by Lineweaver-Burk plot. However, for this to be so, Ph6 should interact with the non-competitive/allosteric domain of the enzyme rather than the active site where docking was reported to have been performed. Hence, how do the authors achieved the docking of Ph6 at the active site contrary to the enzyme inhibition kinetics that was predicted to be non-competitive? This is a key question/concern requiring detailed explanation to support that that the in silico evaluation for Ph6 really makes scientific sense and of significance to the study.

Response:  These lines were added in manuscriptDocking was performed before wet lab experiment and it was found that Ph6 fits in the enzymatic pocket of 2Y9X with an ester carbonyl closer to active site Cu-ions in a similar way as kojic acid does. The described binding mode would allow the substrate to enter in the pocket, providing a useful non-competitive model. However, our docking does not provide significant information to differentiate between non-competitive and competitive inhibition. Figure 4 shows the pose determined by molecular docking for Ph6. Our in vitro enzyme inhibitory assay showed a non-competitive inhibition for Ph6. The Ph6 is an unsubstituted cinnamic acid derivative and does not bear any hydroxyl substituent which is very important in ligand binding through hydrogen bonding, Coulomb and van der Waals interactions and therefore, does not pick any type of interaction with any side chain residues (Fig.4c)”. Please see on page 7, lines 225-235.

  1. Include the interaction plots for Ph6 and Kojic acid as done for Ph5 and Ph9 (Figure 5) and compare the plots for Ph5, Ph6 and Ph9 with that of Kojic acid, being the reference standard used and clearly identify the conserved catalytic residues crucial for tyrosinase activity in consultation with other published works. A mere descriptive manner of presentation as currently done in this study is unacceptable.

Response:

These lines were added in manuscriptEach inhibitor binds this protein in a different manner due to different substitution pattern. The reference inhibitor kojic acid interacts tyrosinase catalytic site with carbonyl group closer to cupper (400,401) ions and side chain hydroxyl is picking up hydrogen bond (1.73Å) with amino acid residue Met280. Please see on page 6, lines 201-204.

To make a comparative study with literature data for the conserved catalytic residues crucial for tyrosinase activity, these lines were added “For the most active compound Ph9 (green in Fig. 4d), the ortho phenolic moiety is interacting through strong H-bond (2.93Å) with side chain carbonyl of Asn260. The para phenolic group is interacting by Cu400 (3.68Å) and Cu401 (3.10Å) and this ring is further stabilized by π-π stacking with His85, His263 and His259. The amide hydrogen is picking up a hydrogen bond (3.36Å) with nitrogen of the neighboring His244. The para methoxy phenyl component in compound Ph9 occupies a distinct area of the binding site picking up a H-bond (3.04Å) between the para methoxy and the hydrogen of side chain Asn81. The most active compound Ph5 from benzoic acids derivatives (Fig. 4b), the para methoxy moiety is forming two strong H-bond interactions with side chain Asn81 (2.07Å) and His85 (2.31Å) respectively and this ring is additionally stabilized by π-π stacking with His85. The amide hydrogen of the inhibitor Ph5 interacts through a hydrogen bond (2.40Å) with nitrogen of the neighboring His244. His259, His263, His296, His61, His85, His94 are conserved residues with Cu (400) and Cu (401) ions in the core region. The residues that played an important role in stabilizing the protein inhibitor complex are Asn260, Asn81, His85, His244 and Met280 and these findings are in accordance with the studies by Pintus et al.,[50]. Please see on page 7, lines 210-225.

Reference:

  1. Pintus, F.; Matos, M.J.; Vilar, S.; Hripcsak, G.; Varela, C.; Uriarte, E.; Santana, L.; Borges, F.; Medda, R.; Di Petrillo, A.; et al. New insights into highly potent tyrosinase inhibitors based on 3-heteroarylcoumarins: Anti-melanogenesis and antioxidant activities, and computational molecular modeling studies. Bioorg. Med. Chem. 2017, 25, 1687–1695. doi: 10.1016/j.bmc.2017.01.037.

  1. PDB: 2Y9X is a crystal structure of tyrosinase in complex with an inhibitor. Define the amino sequence of this structure and include information on how the structure was treated regarding the presence of the inhibitor.

Response: These sentences were addedPDB: 2Y9X is a crystal structure of mushroom tyrosinase in complex with tropolone. Tropolone was removed from active site and all compounds (Ph1-Ph10) were docked in the active site”. Please see page 12, lines 453-455. “His259, His263, His296, His61, His85, His94 are conserved residues with Cu (400) and Cu (401) ions in the core region. Asn81, Asn260, His85, His244, His259 and His263 are the major amino acids involved in catalytic interaction of tyrosinases”. Please see page 7, lines 221-225.

  1. Provide information on the source and purity of the parent/starting compounds for the synthetic routes explored in the study.

Response: This sentence was added in manuscriptAll chemicals used for the synthesis of compounds (Ph1-Ph10) were purchased from Sigma Aldrich (St. Louis, MO, USA)” Please see on page 8, lines 252-253.

  1. English language and style are fine/minor spell check required.

Response: Spellings are thoroughly checked and revised as per reviewer recommendation.

Reviewer 3 Report

Manuscript ID: molecules-1171850

          The article with title “Methoxy substituted tyramine derivatives synthesis, computational studies and tyrosinase inhibitory kinetics” by Zaman Ashraf & Warintorn Ruksiriwanich et al discusses the incorporation of hydroxyl substituted benzoic and cinnamic acid scaffolds into new chemotypes that displayed in vitro inhibitory effect against mushroom and human tyrosinase for the purpose of identifying anti-melanogenic ingredients.

          The most active compounds were 2-((4-methoxyphenethyl)amino)-2-oxoethyl (E)-3- (2,4-dihydroxyphenyl) acrylate (Ph9), with an IC50 of 0.000059 µM, while 2-((4-methoxyphenethyl)amino)-2-oxoethyl cinnamate (Ph6) had an IC50 of 0.0021 µM compared to the positive control, kojic acid IC50 16.7 µM against mushroom tyrosinase. The compound (Ph9) and (Ph6) both exhibited > 90 % inhibitory activity against human tyrosinase inhibitory activity in A375 human melanoma cells compared with 72% for kojic acid. These results showed that compound Ph9 can become a potential candidate for further development of tyrosinase inhibitors.

          The authors have provided a lot of data in this manuscript making it a valuable addition to the scientific literature in the field of tyrosinase inhibitors.

General comment-

          Authors should report the IC50 in abstract as nanomolar, to highlight the results.

          In the introduction authors are comparing extracts Eg Hemp leaf extract, Celastrus  paniculatus seed oil (CPSO), rice bran extract, Ethanolic extracts of medicinal flowers, African marigold (Tagetes erecta L) and their activities against tyrosinase (line 74 onwards till 85). This does not seem reasonable. Comparison should be either of single or isolated natural compounds to highlight the importance of isolation, synthesis and testing protocols for development of anti-tyrosinase drugs. Otherwise the major composition of these extracts should be mentioned to make a point of having cinnamyl esters/amides in them which give bioactivity.

Refer to

  1. A comprehensive review on tyrosinase inhibitors, Journal Of Enzyme Inhibition And Medicinal Chemistry 2019, VOL. 34, NO. 1, 279–309 https://doi.org/10.1080/14756366.2018.1545767

And

  1. An Updated Review of Tyrosinase Inhibitors by Te-Sheng Chang; Int. J. Mol. Sci. 2009, 10(6), 2440-2475; https://doi.org/10.3390/ijms10062440

for updates on natural tyrosinase inhibitors in the introduction section.

          What is the reason to include section 2.1 in the results, is the synthesis of compound 4-methoxyphenyl ethyl chloroacetamide (2) new? If so additional NMR and MS data to be provided for the identification. Section 2.1 to be shifted in materials and methods section; if required the synthesis of target molecules should be written briefly in the results and discussion with yields etc mentioned to highlight the synthetic aspects. In table 1 add a column for the yields of products.

          In the invitro tests, how were the inhibitor applied to the tests? In which solvent were they dissolved is not mentioned in text or in the material/methods section. This is very important. The blank tests were performed with the solvent used to check for its effects?

          In the SAR discussion authors have mentioned the structural aspects relative to the observed bioactivity. Please refer to an elaborate review on resorcinol type substitution in tyrosinase inhibitors for clarity on the observed activity and cite the same if helpful.  An Updated Review of Tyrosinase Inhibitors by Te-Sheng Chang; Int. J. Mol. Sci. 2009, 10(6), 2440-2475; https://doi.org/10.3390/ijms10062440

          In the docking studies the binding energy for Ph6 and Ph9 is in the same range (-7.134 & -7.930 Kcal/mol); while in figure 5 the representation of Ph5 (-5.339 Kcal/mol) and Ph9 is shown. It is important to show the docking for Ph6 also, either in manuscript or as supporting information since the SAR can be discussed/explained in a better way after referencing the above citation- Int. J. Mol. Sci. 2009, 10(6), 2440-2475; In table 3, highlight/ mark as bold the entries and values for Ph6, Ph9 and kojic acid for easy identification.

Moreover, the 1,3-dihydroxy type substitution pattern (seen in Ph 5 and Ph9) governs mixed type inhibition while Ph6 shows non-competitive inhibition. I expect some justification/reasoning to be provided for these results.

          In the introduction at line 96- 98 it was mentioned “Due to the absence of a high resolution crystal structure of the human tyrosinase enzyme, mushroom tyrosinase has frequently served as a model to assess the inhibitory activity of compounds with a fact that both proteins share high sequence similarity in their active core region [44].” This reference is the previous works of the co-authors and not adequate for the protein sequence similarity as mentioned. Authors are expected to justify proper references for similarity of protein sequences between human tyrosinase enzyme, & mushroom tyrosinase and its use for docking. Also refer to doi: 10.1111/pcmr.12225 for additional references on these aspects.

All products are new and the HRMS data is missing. Please provide the spectral data for all products as supporting information file for review and publication.

          Please check the references section and include the available DOI for all the references.

Author Response

Response to reviewer’s comments:

Manuscript ID: Molecules-1171850

Type of Manuscript: Article

Title: Methoxy substituted tyramine derivatives synthesis, computational studies and tyrosinase inhibitory kinetics

Authors: Y. Nazir et al.

REVIEWER 3

  1. Authors should report the IC50 in abstract as nanomolar, to highlight the results.

Response: IC50 values are reported as nanomolar (nM) in abstract and highlighted as per reviewer recommendation. “IC50 of 0.059nM, while 2-((4-methoxyphenethyl)amino)-2-oxoethyl cinnamate (Ph6) had an IC50 of 2.1nM compared to the positive control, kojic acid IC50 16700nM. Enzyme kinetics reflected mixed type of inhibition for inhibitor Ph9 (Ki 0.093nM) and non-competitive inhibition for Ph6 (Ki 2.3nM) revealed from Lineweaver-Burk plots”. Please see on page 1, lines 22,23,26 and 27.

  1. In the introduction authors are comparing extracts Eg. Hemp leaf extract, Celastrus  paniculatus seed oil (CPSO), rice bran extract, Ethanolic extracts of medicinal flowers, African marigold (Tagetes erecta L) and their activities against tyrosinase (line 74 onwards till 85). This does not seem reasonable. Comparison should be either of single or isolated natural compounds to highlight the importance of isolation, synthesis and testing protocols for development of anti-tyrosinase drugs. Otherwise the major composition of these extracts should be mentioned to make a point of having cinnamyl esters/amides in them which give bioactivity.

Response: These sentences have been added in the manuscript as per reviewer’s recommendationsHemp (Cannabis sativa L var. sativa) contains vast number of constituents and the terpenoids and flavonoids present in Hemp extract may be responsible for its tyrosinase inhibitory activity [41]. Tyrosinase inhibition activity of the Hemp leaf extract (IC50 0.049±0.02 mg/ml) is about 9.8 folds lower than standard kojic caid (IC50 0.005 mg/ml)[42]. Celastrus paniculatus seed oil (CPSO) has been reported to exhibit more potent anti-tyrosinase activity (IC50 0.04 mg/ml) than 2-hydroxypropyl β-cyclodextrin inclusion complex CPSO-HPβCD (IC50 0.19 mg/ml) due to the presence of tocopherol [43]. The rice bran extract contains various saturated and unsaturated fatty acids such as palmitic, stearic, linolenic and oleic acids and their ester derivatives. The rice bran loaded in β-cyclodextrin at 1 mg/mL demonstrated the tyrosinase inhibitory activity 1.25 folds of the control and 0.52 folds of the standard theophylline [44]. The key compounds, p-methylanisole, methyl salicylate, 2-phenyl ethyl alcohol, nerolidol, methyl cinnamate, 3-hydroxy-4-phenyl-2-butanoate, cinnamyl alcohol and 2-phenyl ethyl benzoate present in ethanol extract of the medicinal flower (Mimusops elengi L.) gave almost no tyrosinase inhibition activity (IC50 > 200 mg/ml) at all concentrations in comparison with kojic acid (IC50 0.049mg/ml) in the supercritical carbon dioxide fluid extraction (scCO2)[45,46]. Moreover, quercetagetin found in the ethyl acetate (EA) extract of flowers of African marigold (Tagetes erecta L) exhibited tyrosinase inhibitory activity on L-tyrosine (IC50 89.31 µg/ml), higher than α- and β–arbutins (IC50 157.77 and 222.35 µg/ml) and slightly higher (IC50 128.41 µg/ml) than ellagic acid (IC50 151.1 µg/ml) when using L-DOPA as substrate [47]”. Please see on page 2, lines 76-95.

References:

  1. ElSohly, M.A.; Slade, D. Chemical constituents of marijuana: the complex mixture of natural cannabinoids. Life Sci. 2005, 78, 539–548. doi: 10.1016/j.lfs.2005.09.011.
  2. Manosroi, A.; Chankhampan, C.; Kietthanakorn, B.O.; Ruksiriwanich, W.; Chaikul, P.; Boonpisuttinant, K.; Sainakham, M.; Manosroi, W.; Tangjai, T.; Manosroi, J. Pharmaceutical and cosmeceutical biological activities of hemp (Cannabis sativa L. var. sativa) leaf and seed extracts. Chiang Mai J. Sci 2019, 46, 180–195.
  3. Ruksiriwanich, W.; Sirithunyalug, J.; Khantham, C.; Leksomboon, K.; Jantrawut, P. Skin penetration and stability enhancement of celastrus paniculatus seed oil by 2-hydroxypropyl-β-cyclodextrin inclusion complex for cosmeceutical applications. Sci. Pharm. 2018, 86, 33. doi.org/10.3390/scipharm86030033.
  4. Manosroi, A.; Chaikul, P.; Chankhampan, C.; Ruksiriwanich, W.; Manosroi, W.; Manosroi, J. 5-α-Reductase inhibition and melanogenesis induction of the selected Thai plant extracts. Chiang Mai J. Sci. 2018, 45, 220–236.
  5. Wong, K.C.; Teng, Y.E. Volatile components of mimusops elengi L. flowers. J. Essent. Oil Res. 1994, 6, 453–458. doi.org/10.1080/10412905.1994.9698425.
  6. Kietthanakorn, B.; Ruksiriwanich, W.; Manosroi, W.; Manosroi, J.; Manosroi, A. biological activities of supercritical carbon dioxide fluid (scCO2) extracts from medicinal flowers. Chiang Mai J. Sci. 2012, 39, 84–96.
  7. Phrutivorapongkul, A.; Kiattisin, K.; Jantrawut, P.; Chansakaow, S.; Vejabhikul, S.; Leelapornpisid, P. Appraisal of biological activities and identification of phenolic compound of African marigold (Tagetes erecta L) flower extract. Pak. J. Pharm. Sci 2013, 26, 1071–1076.

  1. What is the reason to include section 2.1 in the results, is the synthesis of compound 4-methoxyphenyl ethyl chloroacetamide (2) new? If so additional NMR and MS data to be provided for the identification. Section 2.1 to be shifted in materials and methods section; if required the synthesis of target molecules should be written briefly in the results and discussion with yields etc mentioned to highlight the synthetic aspects. In table 1 add a column for the yields of products.

Response: The intermediate 4-methoxyphenyl ethyl chloroacetamide (2) is a known compound but we synthesized ourselves in the laboratory by following Sidhu et al. This sentence was addedThe intermediate 2-(4-methoxyphenyl)ethan-1-aminechloroacetate (2) was synthesized by following the Sidhu et al., method with some modifications [51]. Please see on page 8, lines 265-266.

Section 2.1 has been shifted to material and methods section under chemistry heading”. Please see page 8, lines 251-263.

An additional column in table 1 for Yields (%) of the target compounds (Ph1-Ph10) as per reviewer’s recommendation has been added. Please see on page 3,4, table 1.

Reference:

  1. Sidhu, G.S.; Sattur, P.B.; Jaleel, S. Synthesis and anticonvulsant activity of some N-phenethylacetamides. J. Pharm. Pharmacol. 1962, 14, 125. doi: 10.1111/j.2042-7158. 1962. tb11065.x.

  1. In the in vitro tests, how were the inhibitor applied to the tests? In which solvent were they dissolved is not mentioned in text or in the material/methods section. This is very important. The blank tests were performed with the solvent used to check for its effects?

Response: These lines were addedAll of the compounds were dissolved in DMSO and blank tests were performed with solvent DMSO. The IC50 values were reported after subtracting the value of DMSO. Kojic acid was used as a reference inhibitor (positive control)”. Please see on page 11, lines 402-404.

  1. In the SAR discussion authors have mentioned the structural aspects relative to the observed bioactivity. Please refer to an elaborate review on resorcinol type substitution in tyrosinase inhibitors for clarity on the observed activity and cite the same if helpful.  An Updated Review of Tyrosinase Inhibitors by Te-Sheng Chang; Int. J. Mol. Sci. 2009, 10(6), 2440-2475;

Response: The suggested review article by Chang et al., was very helpful and has been cited in the SAR section. These lines were addedPhenolics are the largest groups in tyrosinase inhibitors and several polyphenols are accepted as substrates by tyrosinase, depending upon the presence and position of substituent [56]. Cinnamic acid has been reported to possess broad physiological actions involving tyrosinase inhibitory activity [57]. The synthesized compounds (Ph6-Ph10) are all cinnamate esters/amides but differ in the substitution pattern on the cinnamate phenyl ring. Compound (Ph6) bears an unsubstituted cinnamic acid moiety while compound (Ph7), (Ph8) and (Ph9) possess 2-hydroxy, 4-hydroxy and 2,4-dihydroxy substituted cinnamic acid component, respectively. The tyrosinase inhibitory activity of 2,4-dihydroxy cinnamic acid is already reported [58] and its incorporation as an ester/amide in compound Ph9 resulted in an increased inhibitory potential. The replacement of hydrogen from the para position of the cinnamic acid phenyl ring in Ph6 with a 2,4-dihydroxyls in Ph9 led to a significant increase in tyrosinase inhibitory activity (Table 1).

It was found that inhibitor Ph9 bearing dihydroxy substituted cinnamic acid moiety is the most promising compound with an inhibitory potential (IC50 0.00006µM) much better than the standard kojic acid (IC50 16.7µM). The enhanced inhibitory effect of Ph9 is potentially due to the presence of two hydroxyl groups at ortho and para positions of phenyl ring of cinnamic acid moiety. Whereas the Ph6 bearing an unsubstituted cinnamic acid moiety also displayed an excellent enzyme inhibitory activity (IC50 0.00204µM). The inhibitor Ph5 with two hydroxyls at 3,5 positions of benzoic acid moiety also exhibited excellent tyrosinase inhibition (IC50 0.231µM). The bioassay results proved that derivatives with hydroxy substituted cinnamic acid moiety (Ph7-Ph9) displayed greater tyrosinase inhibitory activity compared to hydroxy substituted benzoic acid derivatives (Ph1-Ph5). It has already been demonstrated that phenolic hydroxyls play a key role in tyrosinase inhibition activity as the natural substrates L-tyrosine and L-DOPA also bear the phenolic hydroxyls [59]. Here, we report that the ortho and para-hydroxyls of Ph9 seem to perform a similar role in tyrosinase inhibitory activity. Please see on page 12-13, lines 469-494).

References: 

  1. Chang, T.S. An updated review of tyrosinase inhibitors. Int. J. Mol. Sci. 2009, 10, 2440–2475. doi:10.3390/ijms10062440.
  2. Shi, Y.; Chen, Q.-X.; Wang, Q.; Song, K.-K.; Qiu, L. Inhibitory effects of cinnamic acid and its derivatives on the diphenolase activity of mushroom (Agaricus bisporus) tyrosinase. Food Chem. 2005, 92, 707–712. doi.org/10.1016/j.foodchem.2004.08.031.
  3. Kim, K.-D.; Song, M.-H.; Yum, E.-K.; Jeon, O.-S.; Ju, Y.-W.; Chang, M.-S. 2, 4-dihydroxycinnamic esters as skin depigmenting agents. Bull. Korean Chem. Soc. 2009, 30, 1619–1621. doi.org/10.5012/bkcs.2009.30.7.1619.
  4. Loizzo, M.R.; Tundis, R.; Menichini, F. Natural and synthetic tyrosinase inhibitors as antibrowning agents: an update. Compr. Rev. Food Sci. Food Saf. 2012, 11, 378–398. doi.org/10.1111/j.1541-4337.2012.00191.x

  1. In the docking studies the binding energy for Ph6 and Ph9 is in the same range (-7.134 & -7.930 Kcal/mol); while in figure 5 the representation of Ph5 (-5.339 Kcal/mol) and Ph9 is shown. It is important to show the docking for Ph6 also, either in manuscript or as supporting information since the SAR can be discussed/explained in a better way after referencing the above citation- Int. J. Mol. Sci. 2009, 10(6), 2440-2475; In table 3, highlight/ mark as bold the entries and values for Ph6, Ph9 and kojic acid for easy identification.

Response: Actually, we presented two types of core structures, one with hydroxy substituted benzoic acids (Ph1-Ph5) and others with hydroxy substituted cinnamic acids (Ph7-Ph9). So, we selected one most promising from each series (Ph5, Ph9). But we agree that after adding Ph6 and kojic acid, it will be very clear for the readers to make a good comparison therefore, “figures for Ph6 and kojic acid are added as per reviewer recommendation”. Please see on page 7, figure 4.

“In table 3, suggested entries are marked as bold for the values of Ph6, Ph9 and kojic acid for easy identification as per reviewer recommendation”. Please see on page 7-8, table 3.

The proposed article by the reviewer was helpful in writing SAR. These lines were added in SAR section Phenolics are the largest groups in tyrosinase inhibitors and several polyphenols are accepted as substrates by tyrosinase, depending upon the presence and position of substituent [56]”. Please see on the page 12, lines 469-471.

Reference:

  1. Chang, T.S. An updated review of tyrosinase inhibitors. Int. J. Mol. Sci. 2009, 10, 2440–2475. doi:10.3390/ijms10062440.

  1. Moreover, the 1,3-dihydroxy type substitution pattern (seen in Ph 5 and Ph9) governs mixed type inhibition while Ph6 shows non-competitive inhibition. I expect some justification/reasoning to be provided for these results.

Response: These sentences were added “It is worth mentioning here that the enzyme inhibitory kinetics were performed with a more efficient natural substrate L-DOPA (L-3,4-dihydroxyphenylalanine). Ph5 is 3,5-dihydroxyl substituted benzoic acid derivative and Ph9 is 2,4-dihydroxyl substituted cinnamic acid derivative whereas, Ph6 is an unsubstituted cinnamic acid analogue. As the substrate L-DOPA bears 3,4-dihydroxy substitution this could be the reason that Ph5 and Ph9 bearing hydroxyl substituent gave mixed type inhibition whereas, Ph6 bearing no hydroxyl substituent was non-competitive inhibitor”. Please see on page 4, lines 166-173.

  1. In the introduction at line 96- 98 it was mentioned “Due to the absence of a high resolution crystal structure of the human tyrosinase enzyme, mushroom tyrosinase has frequently served as a model to assess the inhibitory activity of compounds with a fact that both proteins share high sequence similarity in their active core region [44].” This reference is the previous works of the co-authors and not adequate for the protein sequence similarity as mentioned. Authors are expected to justify proper references for similarity of protein sequences between human tyrosinase enzyme, & mushroom tyrosinase and its use for docking. Also refer to doi: 10.1111/pcmr.12225 for additional references on these aspects.

Response: “Proper reference from the work of Lai et al., has been cited as per reviewer recommendation”. Please see on page 3, line108. The suggested article helped in making a nice literature study and these lines were added “Inulavosin and its benzo-derivatives effectively inhibited melanogenesis of the cells [25]”. Please see on page 2, lines 55-56.

Reference:

  1. Fujita, H.; Menezes, J.C.; Santos, S.M.; Yokota, S.; Kamat, S.P.; Cavaleiro, J.A.S.; Motokawa, T.; Kato, T.; Mochizuki, M.; Fujiwara, T.; et al. Inulavosin and its benzo-derivatives, melanogenesis inhibitors, target the copper loading mechanism to the active site of tyrosinase. Pig. cell melan. Res. 2014, 27, 376–386. doi.org/10.1111/pcmr.12225.
  2. Lai, X.; Wichers, H.J.; Soler-Lopez, M.; Dijkstra, B.W. Structure and function of human tyrosinase and tyrosinase-related proteins. Chem. Eur. J. 2018, 24, 47–55. doi: 10.1002/chem.201704410.

  1. All products are new and the HRMS data is missing. Please provide the spectral data for all products as supporting information file for review and publication.

Response: Actually, we performed melting point, FTIR, 1H, 13CNMR and elemental analysis (C, H) and the spectral data has been included in manuscript. Please see on page 9-11, lines 290-393.

Unfortunately, we don’t have HRMS facility in our faculty, but we will consider this for our future studies. The 1H and 13C NMR spectra for all products (Ph1-Ph10) has been added as a supplementary material as per reviewer recommendation. Please see supplementary material file, fig. S1-S10, page 1-10.

  1. Please check the references section and include the available DOI for all the references.

Response: All the references are corrected according to journal format and DOI are added where available as per reviewer recommendations. Please see on page 14-16, lines 530-655.

  1. Moderate English changes required.

Response: The paper has been carefully revised to improve the grammar and readability as per reviewer recommendation.

References:

  1. Solomon, E.I.; Sundaram, U.M.; Machonkin, T.E. Multicopper oxidases and oxygenases. Chem. Rev. 1996, 96(7):2563-606. doi.org/10.1021/cr950046o.
  2. Sánchez-Ferrer, Á.; Neptuno Rodríguez-López, J.; García-Cánovas, F.; García-Carmona, F. Tyrosinase: a comprehensive review of its mechanism. Biochim. Biophys. Acta (BBA)/Protein Struct. Mol. 1995, 1247, 1–11. doi:10.1016/0167-4838(94)00204-T.
  3. Van Holde, K.E.; Miller, K.I.; Decker, H. Hemocyanins and invertebrate evolution. J. Biol. Chem. 2001, 276, 15563–15566. doi:10.1074/jbc.R100010200.
  4. Slominski, A.; Costantino, R. Molecular mechanism of tyrosinase regulation by L-DOPA in hamster melanoma cells. Life Sci. 1991, 48, 2075–2079. doi:10.1016/0024-3205(91)90164-7.
  5. Slominski, A.; Costantino, R. L-tyrosine induces tyrosinase expression via a posttranscriptional mechanism. Experientia 1991, 47, 721–724. doi.org/10.1007/BF01958826.
  6. Slominski, A.; Paus, R. Are L-tyrosine and L-dopa hormone-like bioregulators? J. Theor. Biol. 1990, 143, 123–138. doi: 10.1016/s0022-5193(05)80292-9.
  7. Prota, G. Melanins and melanogenesis; Academic Press, 2012;
  8. Prota, G. The chemistry of melanins and melanogenesis. fortschritte der chemie Org. naturstoffe/progress Chem. Org. Nat. Prod. 1995, 93–148. doi: 10.1007/978-3-7091-9337-2_2.
  9. Kadekaro, A.L.; Kanto, H.; Kavanagh, R.; A-Malek, Z.A. Significance of the melanocortin 1 receptor in regulating human melanocyte pigmentation, proliferation, and survival. Ann. N. Y. Acad. Sci. 2003, 994, 359–365. doi: 10.1111/j.1749-6632.2003.tb03200.x.
  10. Petit, L.; Pierard, G.E. Skin-lightening products revisited. Int. J. Cosmet. Sci. 2003, 25, 169–181. doi.org/10.1046/j.1467-2494.2003.00182.x.
  11. Riley, P.A. Melanogenesis and melanoma. Pig. Cell Res. 2003, 16, 548–552. doi: 10.1034/j.1600-0749.2003.00069.x.
  12. Uong, A.; Zon, L.I. Melanocytes in development and cancer. J. Cell. Physiol. 2010, 222, 38–41. doi: 10.1002/jcp.21935.
  13. Yamaguchi, Y.; Hearing, V.J. Melanocytes and their diseases. Cold Spring Harb. Perspect. Med. 2014, 4, a017046. doi: 10.1101/cshperspect.a017046.
  14. Torihara, M.; Tamai, Y.; Shiono, M.; Tasaka, K. Skin depigmental agent, US4959393, 1990.
  15. Hu, L. Resorcinol derivatives 2000. doi: 10.1007/3-540-28090-1_3.
  16. Shore, L.J.; Rocha, S.A.; McKinney, M.D. Skin lightening agents, compositions and methods, US7270805, 2007.
  17. Joo, Y.H.; Baek, H.S.; Lee, C.S.; Choi, S.J.; Rho, H.S.; Park, M.Y.; Shin, S.S.; Lim, K.M.; Park, Y.H. Benzoic acid amide compound, WO2016159644, 2016.
  18. Okombi, S.; Rival, D.; Boumendjel, A.; Mariotte, A.; Perrier, E. Para-coumaric acid or para-hydroxycinnamic acid derivatives and their use in cosmetic or dermatological compositions, US9089499B2, 2013.
  19. Kaur, S.; Southall, M.D.; Zivin, R.A. Topical application of 1-hydroxyl-3, 5-BIS (4′ hydroxy styryl) benzene, US8846013B2, 2014.
  20. Vanni, A.; Gastaldi, D.; Giunta, G. Kinetic investigations on the double enzymic activity of the tyrosinase mushroom. Ann. Chim. 1990, 80, 35–60.
  21. Takada, R.; Takada, S.; Takeda, K.; Yasumoto, K.; Watanabe, K.; Udono, T.; Saito, H.; Takahashi, K.; Shibahara, S. Induction of melanocyte-specific microphthalmia-associated transcription factor by Wnt-3a. J. Biol. Chem. 2000, 275, 14013–14016. doi: 10.1074/jbc.c000113200.
  22. Bellei, B.; Pitisci, A.; Izzo, E.; Picardo, M. Inhibition of melanogenesis by the pyridinyl imidazole class of compounds: possible involvement of the Wnt/β-catenin signaling pathway. PLoS One 2012, 7, e33021. doi.org/10.1371/journal.pone.0033021.
  23. Horng, C.-T.; Wu, H.-C.; Chiang, N.-N.; Lee, C.-F.; Huang, Y.-S.; Wang, H.-Y.; Yang, J.-S.; Chen, F.-A. Inhibitory effect of burdock leaves on elastase and tyrosinase activity. Exp. Ther. Med. 2017, 14, 3247–3252. doi: 10.3892/etm.2017.4880.
  24. Hałdys, K.; Goldeman, W.; Anger-Góra, N.; Rossowska, J.; Latajka, R. Monosubstituted acetophenone thiosemicarbazones as potent inhibitors of tyrosinase: synthesis, inhibitory studies, and molecular docking. Pharmaceuticals 2021, 14, 74. doi.org/10.3390/ph14010074.
  25. Fujita, H.; Menezes, J.C.; Santos, S.M.; Yokota, S.; Kamat, S.P.; Cavaleiro, J.A.S.; Motokawa, T.; Kato, T.; Mochizuki, M.; Fujiwara, T.; et al. Inulavosin and its benzo-derivatives, melanogenesis inhibitors, target the copper loading mechanism to the active site of tyrosinase. Pig. cell melan. Res. 2014, 27, 376–386. doi.org/10.1111/pcmr.12225.
  26. Tai, A.; Sawano, T.; Yazama, F.; Ito, H. Evaluation of antioxidant activity of vanillin by using multiple antioxidant assays. Biochim. Biophys. Acta (BBA)-General Subj. 2011, 1810, 170–177. doi: 10.1016/j.bbagen.2010.11.004.
  27. Briganti, S.; Camera, E.; Picardo, M. Chemical and instrumental approaches to treat hyperpigmentation. Pig. cell Res. 2003, 16, 101–110. doi: 10.1034/j.1600-0749.2003.00029.x.
  28. Shalit, H.; Dyadyuk, A.; Pappo, D. Selective oxidative phenol coupling by iron catalysis. J. Org. Chem. 2019, 84, 1677–1686. doi.org/10.1021/acs.joc.8b03084.
  29. Rafiq, M.; Nazir, Y.; Ashraf, Z.; Rafique, H.; Afzal, S.; Mumtaz, A.; Hassan, M.; Ali, A.; Afzal, K.; Yousuf, M.R.; et al. Synthesis, computational studies, tyrosinase inhibitory kinetics and antimelanogenic activity of hydroxy substituted 2-[(4-acetylphenyl) amino]-2-oxoethyl derivatives. J. Enzy. Inhib. Med. Chem. 2019, 34, 1562–1572. doi.org/10.1080/14756366.2019.1654468.
  30. Nazir, Y.; Saeed, A.; Rafiq, M.; Afzal, S.; Ali, A.; Latif, M.; Zuegg, J.; Hussein, W.M.; Fercher, C.; Barnard, R.T.; et al. Hydroxyl substituted benzoic acid/cinnamic acid derivatives: Tyrosinase inhibitory kinetics, anti-melanogenic activity and molecular docking studies. Bioorganic Med. Chem. Lett. 2020, 30, 126722. doi.org/10.1016/j.bmcl.2019.126722.
  31. Parvez, S.; Kang, M.; Chung, H.S.; Bae, H. Naturally occurring tyrosinase inhibitors: Mechanism and applications in skin health, cosmetics and agriculture industries. Phyther. Res. 2007. doi: 10.1002/ptr.2184.
  32. Gomez-Cordoves, C.; Bartolome, B.; Vieira, W.; Virador, V.M. Effects of wine phenolics and sorghum tannins on tyrosinase activity and growth of melanoma cells. J. Agric. Food Chem. 2001, 49, 1620–1624. doi: 10.1021/jf001116h.
  33. Park, J.B.; Schoene, N. N-Caffeoyltyramine arrests growth of U937 and Jurkat cells by inhibiting protein tyrosine phosphorylation and inducing caspase-3. Cancer Lett. 2003, 202, 161–171. doi: 10.1016/j.canlet.2003.08.010.
  34. Nesterenko, V.; Putt, K.S.; Hergenrother, P.J. Identification from a combinatorial library of a small molecule that selectively induces apoptosis in cancer cells. J. Am. Chem. Soc. 2003, 125, 14672–14673. doi.org/10.1021/ja038043d.
  35. Phenolic acid amides of phenolic benzylamines against UVA-induced oxidative stress in skin. doi: 10.1046/j.1467-2494.2001.00055.x.
  36. Khan, K.M.; Maharvi, G.M.; Abbaskhan, A.; Hayat, S.; Khan, M.T.H.; Makhmoor, T.; Choudhary, M.I.; Shaheen, F. Three tyrosinase inhibitors and antioxidant compounds from salsola foetida. Helv. Chim. Acta 2003, 86, 457–464. doi.org/10.1002/hlca.200390045.
  37. Son, S.; Lewis, B.A. Free radical scavenging and antioxidative activity of caffeic acid amide and ester analogues: Structure- activity relationship. J. Agric. Food Chem. 2002, 50, 468–472. doi.org/10.1021/jf010830b.
  38. Okombi, S.; Rival, D.; Bonnet, S.; Mariotte, A.-M.; Perrier, E.; Boumendjel, A. Analogues of N-hydroxycinnamoylphenalkylamides as inhibitors of human melanocyte-tyrosinase. Bioorganic Med. Chem. Lett. 2006, 16, 2252–2255. doi: 10.1016/j.bmcl.2006.01.022.
  39. Tamiz, A.P.; Cai, S.X.; Zhou, Z.-L.; Yuen, P.-W.; Schelkun, R.M.; Whittemore, E.R.; Weber, E.; Woodward, R.M.; Keana, J.F.W. Structure- Activity relationship of N-(Phenylalkyl) cinnamides as novel NR2B subtype-selective NMDA receptor antagonists. J. Med. Chem. 1999, 42, 3412–3420. doi: 10.1021/jm990199u.
  40. Roh, J.S.; Han, J.Y.; Kim, J.H.; Hwang, J.K. Inhibitory effects of active compounds isolated from safflower (Carthamus tinctorius L.) seeds for melanogenesis. Biol. Pharm. Bull. 2004, 27, 1976–1978. doi: 10.1248/bpb.27.1976.
  41. ElSohly, M.A.; Slade, D. Chemical constituents of marijuana: the complex mixture of natural cannabinoids. Life Sci. 2005, 78, 539–548. doi: 10.1016/j.lfs.2005.09.011.
  42. Manosroi, A.; Chankhampan, C.; Kietthanakorn, B.O.; Ruksiriwanich, W.; Chaikul, P.; Boonpisuttinant, K.; Sainakham, M.; Manosroi, W.; Tangjai, T.; Manosroi, J. Pharmaceutical and cosmeceutical biological activities of hemp (Cannabis sativa L. var. sativa) leaf and seed extracts. Chiang Mai J. Sci 2019, 46, 180–195.
  43. Ruksiriwanich, W.; Sirithunyalug, J.; Khantham, C.; Leksomboon, K.; Jantrawut, P. Skin penetration and stability enhancement of celastrus paniculatus seed oil by 2-hydroxypropyl-β-cyclodextrin inclusion complex for cosmeceutical applications. Sci. Pharm. 2018, 86, 33. doi.org/10.3390/scipharm86030033.
  44. Manosroi, A.; Chaikul, P.; Chankhampan, C.; Ruksiriwanich, W.; Manosroi, W.; Manosroi, J. 5-α-Reductase inhibition and melanogenesis induction of the selected Thai plant extracts. Chiang Mai J. Sci. 2018, 45, 220–236.
  45. Wong, K.C.; Teng, Y.E. Volatile components of mimusops elengi L. flowers. J. Essent. Oil Res. 1994, 6, 453–458. doi.org/10.1080/10412905.1994.9698425.
  46. Kietthanakorn, B.; Ruksiriwanich, W.; Manosroi, W.; Manosroi, J.; Manosroi, A. Biological activities of supercritical carbon dioxide fluid (scCO2) extracts from medicinal flowers. Chiang Mai J. Sci. 2012, 39, 84–96.
  47. Phrutivorapongkul, A.; Kiattisin, K.; Jantrawut, P.; Chansakaow, S.; Vejabhikul, S.; Leelapornpisid, P. Appraisal of biological activities and identification of phenolic compound of African marigold (Tagetes erecta L) flower extract. Pak. J. Pharm. Sci 2013, 26, 1071–1076.
  48. Lai, X.; Wichers, H.J.; Soler-Lopez, M.; Dijkstra, B.W. Structure and function of human tyrosinase and tyrosinase-related proteins. Chem. Eur. J. 2018, 24, 47–55. doi: 10.1002/chem.201704410.
  49. Baell, J.B.; Nissink, J.W.M. Seven year itch: Pan-assay interference compounds (PAINS) in 2017-utility and limitations. ACS Chem. Biol. 2018, 13, 36–44. doi.org/10.1021/acschembio.7b00903.
  50. Pintus, F.; Matos, M.J.; Vilar, S.; Hripcsak, G.; Varela, C.; Uriarte, E.; Santana, L.; Borges, F.; Medda, R.; Di Petrillo, A.; et al. New insights into highly potent tyrosinase inhibitors based on 3-heteroarylcoumarins: Anti-melanogenesis and antioxidant activities, and computational molecular modeling studies. Bioorg. Med. Chem. 2017, 25, 1687–1695. doi: 10.1016/j.bmc.2017.01.037.
  51. Sidhu, G.S.; Sattur, P.B.; Jaleel, S. Synthesis and anticonvulsant activity of some N-phenethylacetamides. J. Pharm. Pharmacol. 1962, 14, 125. doi: 10.1111/j.2042-7158.1962.tb11065.x.
  52. Seo, W.D.; Ryu, Y.B.; Curtis-Long, M.J.; Lee, C.W.; Ryu, H.W.; Jang, K.C.; Park, K.H. Evaluation of anti-pigmentary effect of synthetic sulfonylamino chalcone. Eur. J. Med. Chem. 2010, 45. doi: 10.1016/j.ejmech.2010.01.049.
  53. Sugimoto, K.; Nishimura, T.; Nomura, K.; Sugimoto, K.; Kuriki, T. Syntheses of arbutin-α-glycosides and a comparison of their inhibitory effects with those of α-arbutin and arbutin on human tyrosinase. Chem. Pharm. Bull. 2003, 51, 798–801. doi: 10.1248/cpb.51.798.
  54. Funayama, M.; Nishino, T.; Hirota, A.; Murao, S.; Takenishi, S.; Nakano, H. Enzymatic synthesis of (+) catechin-α-glucoside and its effect on tyrosinase activity. Biosci. Biotechnol. Biochem. 1993, 57, 1666–1669. doi.org/10.1271/bbb.57.1666.
  55. Mendes, E.; Perry, M. de J.; Francisco, A.P. Design and discovery of mushroom tyrosinase inhibitors and their therapeutic applications. Expert Opin. Drug Discov. 2014, 9, 533–554. doi: 10.1517/17460441.2014.907789.
  56. Chang, T.S. An updated review of tyrosinase inhibitors. Int. J. Mol. Sci. 2009, 10, 2440–2475, doi:10.3390/ijms10062440.
  57. Shi, Y.; Chen, Q.-X.; Wang, Q.; Song, K.-K.; Qiu, L. Inhibitory effects of cinnamic acid and its derivatives on the diphenolase activity of mushroom (Agaricus bisporus) tyrosinase. Food Chem. 2005, 92, 707–712. doi.org/10.1016/j.foodchem.2004.08.031.
  58. Kim, K.-D.; Song, M.-H.; Yum, E.-K.; Jeon, O.-S.; Ju, Y.-W.; Chang, M.-S. 2, 4-dihydroxycinnamic esters as skin depigmenting agents. Bull. Korean Chem. Soc. 2009, 30, 1619–1621. doi.org/10.5012/bkcs.2009.30.7.1619.
  59. Loizzo, M.R.; Tundis, R.; Menichini, F. Natural and synthetic tyrosinase inhibitors as antibrowning agents: an update. Compr. Rev. Food Sci. Food Saf. 2012, 11, 378–398. doi.org/10.1111/j.1541-4337.2012.00191.x.

Round 2

Reviewer 2 Report

The revised manuscript by Nazir et al. significantly addressed most of the concerns previously raised except for the non-alignment between the non-competitiveness of Ph6 in the in vitro assay contrary to docking at the active site of the enzyme as stated by the Authors. Whether docking was performed prior the in vitro experiment or not, there should be clear alignment in the established interactions between Ph6 and the study tyrosinase, and the results of in vitro experimentation.

To this end, it is suggested that Ph6 be docked at other regulatory site(s) of the enzyme other than the active site and it is only then that the docking results would make scientific sense.  

Author Response

Response to reviewer’s comments (Round-2):

Manuscript ID: Molecules-1171850

Type of Manuscript: Article

Title: Methoxy substituted tyramine derivatives synthesis, computational studies and tyrosinase inhibitory kinetics

Authors: Y. Nazir et al.

REVIEWER 2

  1. The revised manuscript by Nazir et al. significantly addressed most of the concerns previously raised except for the non-alignment between the non-competitiveness of Ph6 in the in vitro assay contrary to docking at the active site of the enzyme as stated by the Authors. Whether docking was performed prior the in vitro experiment or not, there should be clear alignment in the established interactions between Ph6 and the study tyrosinase, and the results of in vitro experimentation.
  2. To this end, it is suggested that Ph6 be docked at other regulatory site(s) of the enzyme other than the active site and it is only then that the docking results would make scientific sense.  

Response:  These lines were added as per reviewer recommendationThe large size and planner shape of Ph6 prevents it from entering the narrow binuclear copper-binding site and bounds to a shallow area at the surface of the enzyme pocket suggesting a non-competitive inhibition. However, a hydrogen bond (2.31Å) is observed between the amide carbonyl and side chain Asn81. Both para methoxy phenyl and cinnamic acid phenyl groups form π-π interaction with residues His85 and Phe192, respectively. Our in vitro enzyme inhibitory assay showed a non-competitive inhibition for Ph6. The Ph6 is an unsubstituted cinnamic acid derivative and does not bear any hydroxyl substituent which is very important in ligand binding through hydrogen bonding, Coulomb and van der Waals interactions (Fig.4c)”. Please see on page 7, lines 226-236. A new plot for inhibitor Ph6 has been incorporated in figure 4. Please see on page 7.